# Transmission and diversity of *Schistosoma haematobium* and *S. bovis* and their freshwater intermediate snail hosts *Bulinus globosus* and *B. nasutus* in the Zanzibar Archipelago, United Republic of Tanzania

**Tom Pennance**[1,2,3,4] *, **Shaali Makame Ame**[5], **Amour Khamis Amour**[5], **Khamis Rashid Suleiman**[5], **Mtumweni Ali Muhsin**[6], **Fatma Kabole**[6], **Said Mohammed Ali**[5], **John Archer**[1,3], **Fiona Allan**[1,3,7], **Aidan Emery**[1,3], **Muriel Rabone**[1,3], **Stefanie Knopp**[8,9], **David Rollinson**[1,3], **Joanne Cable**[2], **Bonnie L. Webster**[1,3]

**1** Department of Science, Natural History Museum, London, United Kingdom, **2** School of Biosciences, Cardiff University, Cardiff, United Kingdom, **3** London Centre for Neglected Tropical Disease Research, London, United Kingdom, **4** Department of Basic Medical Sciences, College of Osteopathic Medicine of the Pacific–Northwest, Western University of Health Sciences, Lebanon, Oregon, United States of America, **5** Public Health Laboratory-Ivo de Carneri, Pemba, United Republic of Tanzania, **6** Neglected Diseases Program, Ministry of Health Zanzibar, United Republic of Tanzania, **7** The Scottish Oceans Institute, Gatty Marine Laboratory, University of St Andrews, East Sands, St Andrews, United Kingdom, **8** Swiss Tropical and Public Health Institute, Allschwil, Switzerland, **9** University of Basel, Basel, Switzerland

* t.pennance@nhm.ac.uk

## Abstract

### Background

The Zanzibar Archipelago (Pemba and Unguja islands) is targeted for the elimination of human urogenital schistosomiasis caused by infection with *Schistosoma haematobium* where the intermediate snail host is *Bulinus globosus*. Following multiple studies, it has remained unclear if *B. nasutus* (a snail species that occupies geographically distinct regions on the Archipelago) is involved in *S. haematobium* transmission on Zanzibar. Additionally, *S. haematobium* was thought to be the only *Schistosoma* species present on the Zanzibar Archipelago until the sympatric transmission of *S. bovis*, a parasite of ruminants, was recently identified. Here we re-assess the epidemiology of schistosomiasis on Pemba and Unguja together with the role and genetic diversity of the *Bulinus* spp. involved in transmission.

### Methodology/Principal findings

Malacological and parasitological surveys were conducted between 2016 and 2019. In total, 11,116 *Bulinus* spp. snails were collected from 65 of 112 freshwater bodies surveyed. *Bulinus* species identification were determined using mitochondrial *cox*1 sequences for a representative subset of collected *Bulinus* (n = 504) and together with archived museum specimens (n = 6), 433 *B. globosus* and 77 *B. nasutus* were identified. Phylogenetic analysis

**Data Availability Statement:** All DNA sequence files are available from the GenBank database (accession numbers MT380520-MT380560). All other relevant data are within the manuscript and its Supporting Information files.

**Funding:** The study was partially funded by a Wellcome Trust Seed Award (https://wellcome.org) grant number 207728 (awarded to BLW). TP was funded by the NERC GW4+ DTP (https://www.nercgw4plus.ac.uk) and the Natural Environmental Research Council (https://www.nerc.com), number NE/L002434/1. Data and samples from the ZEST project were also used in the current study, ZEST was funded by the University of Georgia Research Foundation Inc., which is funded by the Bill & Melinda Gates Foundation (https://www.gatesfoundation.org/) for the Schistosomiasis Consortium for Operational Research and Evaluation (SCORE; https://score.uga.edu/) projects (prime award no. 50816, subaward no. RR374-053/4893206 to DR). FA, AE and MR were funded by the Wellcome Trust (https://wellcome.org), for the SCAN: Schistosomiasis Collection at the Natural History Museum, grant number 104958/Z/14/Z, in which many of the ZEST samples were accessioned. The funders had no role in study design, data collection and analysis, decision to publish, or preparation of the manuscript.

**Competing interests:** The authors have declared that no competing interests exist.

of *cox*1 haplotypes revealed three distinct populations of *B. globosus*, two with an overlapping distribution on Pemba and one on Unguja. For *B. nasutus*, only a single clade with matching haplotypes was observed across the islands and included reference sequences from Kenya. *Schistosoma haematobium* cercariae (n = 158) were identified from 12 infected *B. globosus* and one *B. nasutus* collected between 2016 and 2019 in Pemba, and cercariae originating from 69 *Bulinus* spp. archived in museum collections. *Schistosoma bovis* cercariae (n = 21) were identified from seven additional *B. globosus* collected between 2016 and 2019 in Pemba. By analysing a partial mitochondrial *cox*1 region and the nuclear ITS (1–5.8S-2) rDNA region of *Schistosoma* cercariae, we identified 18 *S. haematobium* and three *S. bovis* haplotypes representing populations associated with mainland Africa and the Indian Ocean Islands (Zanzibar, Madagascar, Mauritius and Mafia).

## Conclusions/Significance

The individual *B. nasutus* on Pemba infected with *S. haematobium* demonstrates that *B. nasutus* could also play a role in the local transmission of *S. haematobium*. We provide preliminary evidence that intraspecific variability of *S. haematobium* on Pemba may increase the transmission potential of *S. haematobium* locally due to the expanded intermediate host range, and that the presence of *S. bovis* complicates the environmental surveillance of schistosome infections.

### Author summary

Schistosomiasis is a snail-borne neglected tropical disease caused by parasitic blood flukes of the genus *Schistosoma*. Human urogenital schistosomiasis is targeted for elimination on the Zanzibar Archipelago, United Republic of Tanzania, with multiple interventions being implemented to curtail transmission of the parasite to humans on the islands since 2012. Environmental surveillance for schistosomiasis transmission by collecting intermediate host snails, checking snails for *Schistosoma* infection, and preserving collected snails and *Schistosoma* parasites offers the possibility for molecular analyses to investigate the evolutionary/genetic relationships of both snails and parasites. Schistosome transmission on Zanzibar was believed to involve a single schistosome species (*Schistosoma haematobium*) transmitted via a single intermediate host species (*Bulinus globosus*). However, our findings demonstrate the locally established presence of *S. bovis*, responsible for bovine intestinal schistosomiasis, and an extended intermediate host compatibility of *S. haematobium* with the snail *B. nasutus* on Pemba. Increased parasite diversity and intermediate host species compatibility may increase the transmission of *Schistosoma* species on Zanzibar and stretch resources for public health interventions with the need for *Schistosoma* species specific surveillance.

## Introduction

Schistosomiasis is a snail-borne neglected tropical disease (NTD), that can cause severe morbidity and mortality in both humans and animals [1,2]. *Schistosoma haematobium* and *S. mansoni* are the two species responsible for most cases of human schistosomiasis in sub-Saharan Africa, causing urogenital and intestinal schistosomiasis, respectively. Community or school-

based treatment of schistosomiasis using the only recommended preventive chemotherapeutic drug currently available, praziquantel (Merck KGaA), is the most common and effective means of alleviating disease burden [3,4]. As prevalence of the infection in humans moves towards elimination in parts of sub-Saharan Africa following large-scale multi-year treatment programmes, it is apparent that low-level transmission is continuing, facilitated by the presence of freshwater snail intermediate hosts enabling reinfection [5,6]. Monitoring schistosome infections in snails as part of control and elimination surveillance, either supplementing human/veterinary parasitology surveys or as a stand-alone measure, will aid in establishing whether *Schistosoma* spp. transmission is persisting, interrupted or re-established following elimination.

Unguja and Pemba islands, collectively known as Zanzibar (United Republic of Tanzania), are endemic for urogenital schistosomiasis. Zanzibar has a long history of pioneering urogenital schistosomiasis research and control dating back to the 1920s. This ranges from investigating the freshwater snails involved in disease transmission to early trials of schistosomacidal drugs and assessing disease prevalence through low cost diagnostics [7–15]. More recently, the islands have been targeted for elimination with concerted efforts within the Zanzibar Elimination of Schistosomiasis Transmission (ZEST) project (2012–2017). This trialled the additive impact of integrated interventions (such as mollusciciding against the snail intermediate host or educational measures for behavioural change) in combination with bi-annual mass drug administration (MDA) and MDA alone [16–19]. Over the course of the ZEST project, prevalence was significantly reduced across the islands, but transmission was not interrupted [18], leaving focal endemicity in hotspot areas that require new methods of surveillance and tailored interventions [19–21]

Of the four endemic snail species of *Bulinus* on Zanzibar (*B. globosus*, *B. nasutus*, and two *B. forskalii* group taxa: *B. forskalii*; and a taxon currently undescribed and presented previously as *Bulinus* sp.), it has been concluded from earlier studies investigating local intermediate host compatibility that only *B. globosus* is a compatible intermediate host involved in *S. haematobium* transmission, with *B. nasutus* being refractory to *S. haematobium* infection on Zanzibar [22–25]. *Schistosoma haematobium* transmission in the past couple of decades was therefore considered to be restricted across the islands to only freshwater bodies where *B. globosus* resided (Fig 1), with the distribution of *B. globosus* on the islands also being constrained by this species' habitat preferences such as water hardness [23]. Therefore, freshwater habitats (e.g., in the southern part of Unguja) previously identified as containing only *B. nasutus* (a freshwater snail species closely related in the *Bulinus africanus* species group and morphologically overlapping with *B. globosus*) were considered free of *S. haematobium* transmission (Fig 1), despite there being evidence that *B. nasutus* can harbour pre-patent *Schistosoma* spp. on Pemba [26]. *Schistosoma haematobium* populations on Zanzibar are considered more genetically diverse in comparison to mainland African populations, with the presence of both Group 1 (mainland Africa) and the more diverse Group 2 (Indian Ocean Islands) strains [27,28]. However, it was not until recently that a second *Schistosoma* species, *S. bovis* which infects ruminants, was identified as being transmitted by *B. globosus* on Pemba [29]. Surveillance of *Schistosoma* transmission in Zanzibar is therefore complicated by not only the presence of two morphologically indistinguishable larval *Schistosoma* species shed from *Bulinus* snails, but also two morphologically overlapping *Bulinus* species.

Despite several studies demonstrating the incompatibility of *S. haematobium* with *B. nasutus* on Unguja [22–25], overlapping areas of *B. nasutus* and urogenital schistosomiasis endemicity have been shown to exist predominantly on Pemba (Fig 1), and reports up until 1962 demonstrate high *S. haematobium* prevalence and only *B. nasutus* presence in the south of Unguja [9,10,23]. This suggests that *B. nasutus* could be acting as an intermediate host for *S.*

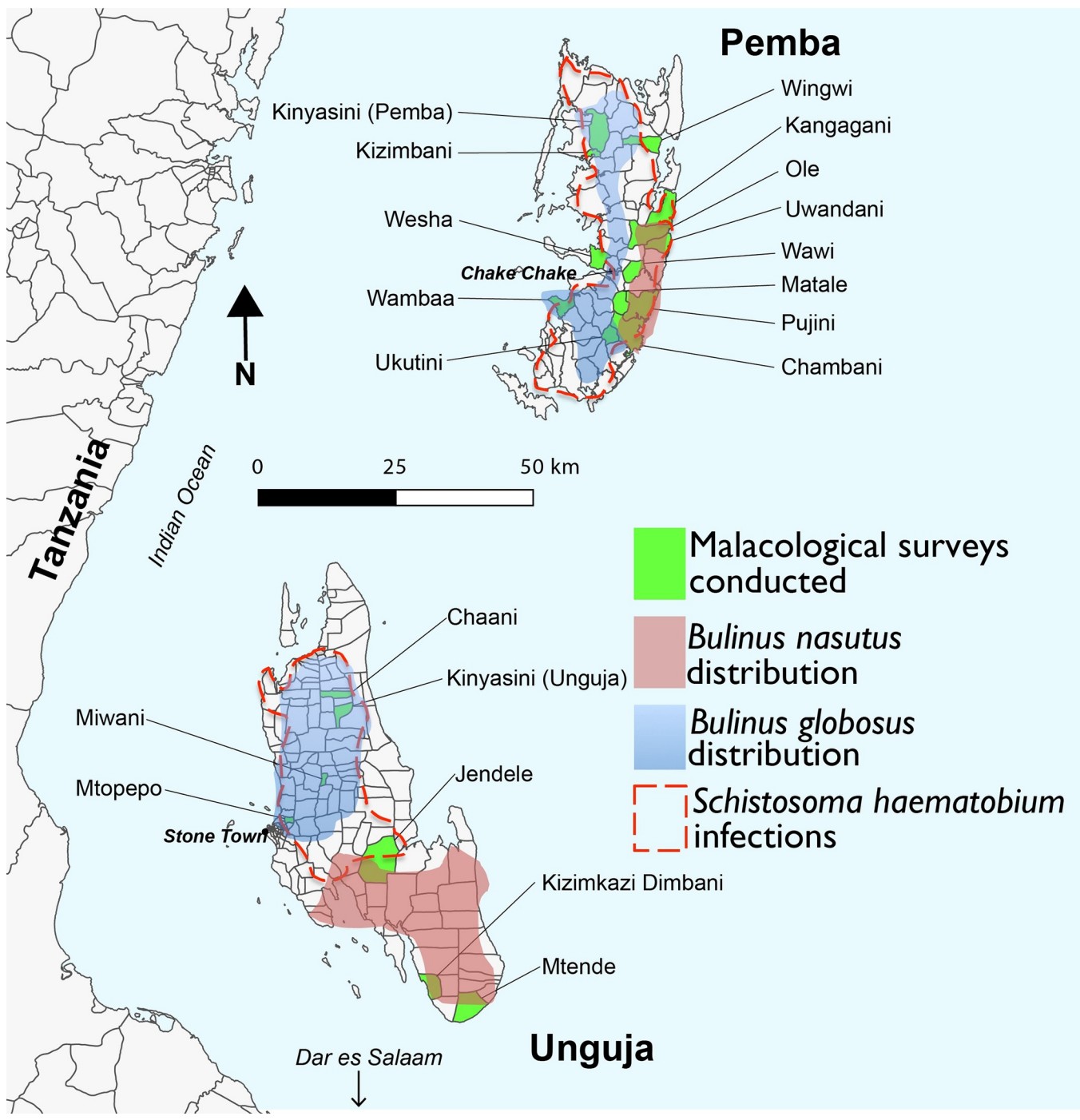

**Fig 1. Map of Pemba and Unguja islands (Zanzibar, United Republic of Tanzania) showing shehias where malacological surveys for *Bulinus* species were conducted in the current study and predicted distributions of *Bulinus* spp. and *Schistosoma haematobium* endemicity based on previous findings.** *Bulinus* spp. distribution inferred from Stothard *et al.* [23] and Pennance *et al.* [29,30]. *Schistosoma haematobium* infection distribution interpreted from Knopp *et al.* [18]. Digital shape files for Unguja and Pemba administrative regions were obtained from DIVA-GIS (https://www.diva-gis.org).

*haematobium* on Pemba and in the past on Unguja, as is the case in nearby coastal regions of Kenya [31] and Tanzania [32–37], with the addition of the closely related snail species *B. (nasutus) productus* also identified as involved in transmission in mainland Tanzania [38].

In this study we aimed to investigate the current transmission status of urogenital and bovine schistosomiasis on Pemba and Unguja islands by monitoring the species distributions and genetic diversity of *B. globosus* and *B. nasutus* and their associated *Schistosoma* spp. collected between 2005 and 2019. Inferences are made on how the findings may impact schistosomiasis control, surveillance and elimination in Zanzibar.

## Methods

### Ethics statement

Ethical approval for the collection and analyses of the snail and schistosome samples collected during the ZEST project were obtained from the Zanzibar Medical Research Ethics Committee in Zanzibar, United Republic of Tanzania (ZAMREC, reference no. ZAMREC 0003/Sept/011), the "Ethikkommission beiber Basel" (EKBB) in Basel, Switzerland (reference no. 236/11) and the Institutional Review Board of the University of Georgia in Athens, Georgia, United States of America (project no. 2012-10138-0) [18]. The ZEST study is registered with the International Standard Randomized Controlled Trial Number register (ISRCTN48837681). Additional sampling and analyses of snails in Unguja and Pemba were conducted in agreement with the Neglected Diseases Program of the Zanzibar Ministry of Health and the Public Health Laboratory-Ivo de Carneri respectively. All other snail and cercariae samples used were accessioned in the Schistosomiasis Collection at the Natural History Museum [39].

### Sampling of *Bulinus* spp. on Pemba and Unguja islands

Following oral approval to conduct surveys by local Shehas, community leaders that locally govern each area (Shehia), human-freshwater contact sites were located from previous reports or with the help of local residents. Coordinates were taken at 112 freshwater sites (109 on Pemba and three on Unguja) across 20 shehias (Fig 1) using a Garmin GPSMAP 62sc device (Garmin, Kansas City, USA) and each water body was surveyed for the presence of intermediate host snails. Between 1 and 5 surveys were conducted at each human-freshwater contact site across Pemba and Unguja in October 2016, October 2017, February, July and November 2018 and January 2019 (Table 1).

At each site, snails were identified using shell morphology to their genera, with any non-*Bulinus africanus* group snails (namely *B. forskalii* group snails) being returned to the collection site. Snails were collected by hand predominantly from submerged vegetation and tree roots around the water's edge that were in close proximity to access points to the water. Each site was surveyed for 15 minutes by three collectors, starting from access points and then searching as much of the accessible perimeter of the waterbody during this time. Snails morphologically identified as either *B. globosus* or *B. nasutus* (species differentiation on morphology alone not possible) were placed in collection pots and transported back to either the Public Health Laboratory-Ivo de Carneri (Chake Chake, Pemba) or the Neglected Diseases Program laboratory (Zanzibar Town, Unguja) where they were counted and housed in plastic trays with bottled water and covered by a glass lid overnight to acclimatise. The following morning (before 08:00), snails were rinsed with bottled water (to mitigate carry over of any cercariae between snails) and examined for cercarial shedding by placing individuals in wells of 12-well ELISA plates filled to approximately two thirds with bottled water and placed under indirect sunlight. Each well was checked using a dissection microscope after two hours and again eight hours after first sunlight to capture schistosomes with different shedding patterns [40]. An experienced microscopist distinguished furcocercous schistosome cercariae from other species using descriptions of *Schistosoma* spp. under a dissecting microscope [41]; a subset of any shed cercariae were individually captured and pipetted in 3.5 μl onto Whatman FTA cards

**Table 1.** *Bulinus* spp. collected in Pemba and Unguja islands (Zanzibar, United Republic of Tanzania) and samples analysed from previous collections and curated within the Schistosomiasis Collection at the Natural History Museum (SCAN).

| Island Shehia | Collection Dates | No. freshwater bodies surveyed | No. *Bulinus* collected [a] | No. *Bulinus* molecularly identified | *Bulinus* spp.[c] of the subset molecularly identified | No. *Bulinus* snails infected with *S. h* (*S. b*) [e] |
|---|---|---|---|---|---|---|
| **Pemba** | | | | | | |
| Ukutini | 21/02/2018 | 10 | 750 | 29 | *B. globosus* | 0 |
| | 18/07/2018 | 10 | 996 | 29 | *B. globosus* | 0 |
| | 19/01/2018 | 10 | 724 | 23 | *B. globosus* | 0 |
| Pujini | 11/10/2017 | 10 | 210 | 20 | *B. nasutus* | 0 |
| | 15/02/2018 | 10 | 268 | 17 | *B. nasutus* | 0 |
| | 22/07/2018 | 10 | 467 | 19 | *B. globosus* & *B. nasutus* | 0 |
| | 19/11/2018 | 10 | 0 | 0 | NC [b] | 0 |
| Kizimbani | 25/10/2016 | 3 | 199 | 6 | *B. globosus* | 0 |
| | 10/10/2017 | 3 | 289 | 9 | *B. globosus* | 2 |
| | 16/02/2018 | 4 | 283 | 14 | *B. globosus* | 0 |
| | 20/07/2018 | 4 | 102 | 12 | *B. globosus* | 0 |
| | 22/11/2018 | 4 | 75 | 11 | *B. globosus* | 0 |
| Kinyasini | 20/10/2016 | 9 | 463 | 17 | *B. globosus* | 1 (5) |
| | 11/10/2017 | 11 | 744 | 27 | *B. globosus* | 0 |
| | 14/10/2018 | 11 | 627 | 31 | *B. globosus* | 7 |
| | 19/07/2018 | 11 | 369 | 25 | *B. globosus* | 0 |
| | 22/11/2018 | 11 | 241 | 25 | *B. globosus* | 1 (2) |
| Wambaa | 26/10/2016 | 8 | 79 | 3 | *B. globosus* | 0 |
| | 08/10/2017 | 9 | 1033 | 24 | *B. globosus* | 0 |
| | 19/02/2018 | 10 | 267 | 25 | *B. globosus* | 0 |
| | 21/07/2018 | 11 | 356 | 18 | *B. globosus* | 0 |
| | 21/11/2018 | 11 | 348 | 22 | *B. globosus* | 0 |
| Wawi | 22/10/2016 | 2 | 149 | 3 | *B. globosus* | 0 |
| | 05/10/2017 | 3 | 183 | 7 | *B. globosus* | 0 |
| | 13/02/2018 | 3 | 10 | 4 | *B. globosus* | 0 |
| | 17/07/2018 | 3 | 1 | 1 | *B. globosus* | 0 |
| | 20/11/2018 | 3 | 4 | 3 | *B. globosus* | 0 |
| Ole | 27/10/2016 | 11 | 0 | 0 | NC [b] | 0 |
| | 06/10/2017 | 13 | 4 | 3 | *B. globosus* | 0 |
| | 12/02/2018 | 13 | 3 | 3 | *B. globosus* | 0 |
| | 17/07/2018 | 13 | 0 | 0 | NC [b] | 0 |
| | 23/11/2018 | 13 | 0 | 0 | NC [b] | 0 |
| Matale | 27/10/2016 | 10 | 0 | 0 | NC [b] | 0 |
| | 09/10/2017 | 11 | 420 | 16 | *B. globosus* | 0 |
| | 19/02/2018 | 11 | 0 | 0 | NC [b] | 0 |
| | 23/07/2018 | 11 | 212 | 16 | *B. globosus* | 0 |
| | 20/11/2018 | 11 | 167 | 14 | *B. globosus* | 0 |
| Chambani | 24/10/2016 | 9 | 291 | 10 | *B. globosus* & *B. nasutus* | 1 |
| | 07/10/2017 | 13 | 91 | 0 | NC [b] | 0 |
| | 09/02/2018 | 13 | 0 | 0 | NC [b] | 0 |
| | 24/07/2018 | 2 | 16 | 0 | NC [b] | 0 |
| | 19/11/2018 | 2 | 16 | 0 | NC [b] | 0 |
| Uwandani | 19/10/2016 | 9 | 136 | 10 | *B. nasutus* | 0 |
| | 05/10/2017 | 9 | 261 | 1 | *B. nasutus* | 0 |
| | 20/02/2018 | 10 | 18 | 0 | NC [b] | 0 |

*(Continued)*

**Table 1.** (Continued)

| Island Shehia | Collection Dates | No. freshwater bodies surveyed | No. *Bulinus* collected [a] | No. *Bulinus* molecularly identified | *Bulinus* spp.[c] of the subset molecularly identified | No. *Bulinus* snails infected with *S. h (S. b)* [e] |
|---|---|---|---|---|---|---|
| **Pemba** | | | | | | |
| | 17/07/2018 | 10 | 28 | 0 | NC[b] | 0 |
| | 26/11/2018 | 2 | 12 | 0 | NC[b] | 0 |
| Kangagani | 22/01/2019 | 1 | 198 | 1 | *B. nasutus* | 1 |
| Wingwi | - | 1[d] | NA[a] | 1 | *B. globosus* | 0 |
| Wesha | 22/10/16 | 6 | 0 | 0 | NC[b] | 0 |
| **Unguja** | | | | | | |
| Jendele | - | 1[d] | NA[a] | 1 | *B. nasutus* | 0 |
| Miwani | 06/02/2013 | 1[d] | NA[a] | 1 | *B. globosus* | 1 |
| Kinyasini | 10/10/2016 | 1[d] | NA[a] | 1 | *B. globosus* | 1 |
| Mtopepo | - | 1[d] | NA[a] | 1 | *B. nasutus* | 0 |
| Chaani | 06/07/2005 | 1[d] | NA[a] | 1 | *B. globosus* | 0 |
| Mtende | 17/07/2018 | 1 | 4 | 4 | *B. nasutus* | 0 |
| Kizimkazi Dimbani | 23/07/2018 | 1 | 2 | 2 | *B. nasutus* | 0 |

[a] NA indicates that data for the total number of *Bulinus* snails collected during the survey were not recorded.

[b] NC indicates that no snails were either collected or identified during these surveys.

[c] Species inferred from *cox*1 similarity to reference sequences.

[d] Snails from the SCAN collections.

[e] *S. h* = *S. haematobium*, *S. b* = *S. bovis* inferred from *cox*1 and complete ITS (1–5.8S-2) rDNA region similarity to reference sequences. All *Bulinus* snails with patent (shedding schistosome cercariae) infections, and schistosome DNA extracted from cercariae.

(Whatman, Part of GE Healthcare, Florham Park, USA) for molecular characterisation. All snails were preserved in 100% ethanol for subsequent molecular characterisation as previously described (see [29]).

A targeted malacological survey was conducted in late January 2019 to collect *B. nasutus* snails at one site in Kangagani on Pemba previously identified as inhabited by *B. nasutus* (see [26]) (Fig 1). These snails were maintained in laboratory aquaria (dimensions 45x30x30cm, 40.5L, filled to approximately two thirds full with water from the collection site and equipped with an air pump for continuous aeration) at a maximum density of 100 *Bulinus* individuals per aquarium. Snails were re-checked for shedding of schistosome cercariae three weeks later in an effort to capture any infections that may have been pre-patent during the initial screen. Aquarium water was replaced twice a week using water from the site of collection (Kangagani). Water was only used in aquaria after storage for at least 48 hours in transparent 15L water containers, allowing for any sediment to settle and eliminate the risk of introducing live schistosome eggs, miracidia or cercariae into the aquaria. Snails were fed on dried lettuce when all lettuce in the tank had been eaten. Dead snails were removed from the aquaria daily.

### Archived samples included in the analysis

*Bulinus* samples, from Unguja and Pemba, accessioned within the Schistosomiasis Collection at the Natural History Museum (SCAN) [39] were included in the study providing material from areas not covered in the malacological surveys described above. Samples were only included if associated geographical information was available. Six snails were included, five from Unguja (Jendele, Miwani, Kinyasini, Mtopepo, Chaani) and one from Pemba (Wingwi) as presented in Table 1. Two of these snails from Miwani and Kinyasini on Unguja were

recorded as patent with *S. haematobium* when they were collected, as this was later confirmed by cercarial molecular analysis as described below. The remaining four *Bulinus* snails from the SCAN collection were negative for patent *Schistosoma* infections (S1 Table). One of the infected *B. globosus* identified in the SCAN repository (MCF389B0F0286, S1 Table) could not be associated with its emerging *S. haematobium* cercariae as it was preserved together with two other *B. globosus* also infected with *S. haematobium* from the same site.

## *Bulinus* spp. molecular characterisation

Since a significant degree of morphological overlap exists between *B. globosus* and *B. nasutus*, a molecular marker was used to unequivocally identify a subset of the *Bulinus* snails collected (n = 504), and those taken from archived specimens (n = 6). The shell was removed by crushing and the use of sterile forceps from the preserved sample and gDNA from whole snail tissue was extracted using either the Qiagen BioSprint 96 DNA Blood Kit following manufacturer's instructions (Qiagen, Manchester, UK) or the Qiagen DNeasy Blood & Tissue Kit modified protocol (Qiagen, Manchester, UK) using double volumes of the lysis buffers [29]. Since not all snails could be identified using molecular characterisation due to cost and time constraints, a minimum of three non-patent snails per site per malacological survey (except for those collected on Pemba during October 2016 and from Chambani and Uwandani in 2017/2018) were randomly selected for identification. Molecular characterisation was performed for all snails with patent *Schistosoma* infections and snails retrieved from SCAN.

DNA was extracted from a total of 510 *Bulinus* spp. snails, from Unguja (11 snails collected from 8 sites) and Pemba (499 snails collected from 63 sites). A 623 bp partial region of mitochondrial *cox*1 DNA was amplified and Sanger sequenced following previously described methods (see [29]). Sanger sequence data were edited, manually trimmed to 463–621 bp and aligned in Sequencher v5.4.6 (GeneCodes Corp., Michigan, USA) before being collapsed into *cox*1 haplotype groups. Species identification were confirmed by alignment and phylogenetic analysis (see below) of *Bulinus cox*1 haplotypes, as presented in S2 Table, with reference data for *B. globosus* and *B. nasutus* [42].

## *Schistosoma* spp. cercariae molecular characterisation

From each infected snail either two or six *Schistosoma* cercariae were processed individually for molecular identification. Six cercariae were individually processed from each infected snail collected between 2016 and 2019 (n = 20 snails) and two cercariae for snails collected as part of the ZEST study (n = 69 snails) [16–18] and made available via SCAN (S3 Table). Following elution of parasite DNA from Whatman FTA cards [43], *Schistosoma* species identification was confirmed by mito-nuclear genetic profiling targeting the partial mitochondrial *cox*1 region (956 bp) and the complete nuclear ITS (1–5.8S-2) rDNA region (967 bp) from each individual cercariae as described in [27,28]. Both mitochondrial and nuclear DNA were analysed for species identification and to identify any hybridisation [44]. The *cox*1 data were also used for genetic diversity and phylogenetic analyses. The sequence data were manually edited and trimmed to 750 bp for *cox*1, and 880bp for ITS, using Sequencher v5.4.6 (GeneCodes Corp., Michingan, USA). The *cox*1 species identity was confirmed by comparison to nucleotide sequences using NCBI-BLAST [45] and the ITS species ID was confirmed by comparison to reference data as described [28,29]. Species identification of each cercariae was confirmed by concordance between the *cox*1 and ITS genetic profiles. Cercariae of identical *cox*1 sequences were collapsed into *cox*1 haplotype groups for further phylogenetic analysis (S4 Table). ITS data were not used for phylogenetic analysis since no intra-species diversity was observed.

### Phylogenetic *cox*1 analysis of *Bulinus* spp. and *Schistosoma* spp

The *Bulinus* haplotype data were imported into Geneious v11.1.4 [46] for phylogenetic analysis together with reference data for *B. nasutus* and *B. globosus* collected previously from East Africa (Zanzibar, Tanzania, Kenya, Uganda, Mafia Island; [42] and an outgroup of *Biomphalaria glabrata* available from GenBank (Accession: NC005439) [47]. Haplotype alignments were performed using ClustalW v2.1 [48] executed in PAUP* [49] and then an appropriate evolutionary nucleotide substitution model (HKY + I + G; -lnl 2020.2144, AIC 4052.4287) was selected in MrModelTest v2.4 [50] using the Akaike Information Criterion. Bayesian inference was performed using MrBayes v3.2.7a [51]. The burn-in was set at 3.5 million generations for consistency after confirming that the average standard deviation of split frequencies (ASDOSF) reported from MrBayes output was at least <0.01 by this point. Clades were considered to have high nodal support if Bayesian inference posterior probability was ≥0.95; tree nodes with <0.95 were collapsed in SumTrees v4.4.0 [52].

*Schistosoma* cercarial haplotype phylogenetic analyses were performed as above with *S. curassoni* (AY157210; [53]) as the outgroup. Analyses also included published *S. haematobium* haplotypes from Zanzibar (GU257334 –GU257360; [28]), and *S. bovis* from Pemba (MH014042 & MH014043; [29] and OK484569 [54]), mainland Tanzania (AY157212; [53]) and Cameroon (MH647141; [55]), with the alignment trimmed to 750 bp to maintain uniform ends. A Templeton, Crandall and Sing's (TCS) haplotype network analysis was also conducted using PopART [56,57] using the same sequence alignment.

### Statistical analysis

Within the molecular analyses *S. haematobium* cercariae were assigned to either the Group 1 or Group 2 *cox*1 haplotype group [27], and Chi-squared tests were performed in R v.4.0.0 [58] to investigate any differences in the abundance of the two groups in relation to their snail host species and geographical distribution.

### Spatial distribution of *Bulinus* and *Schistosoma* species

*Bulinus* and *Schistosoma* spp. distribution data were visualised using QGIS v3.0.1 Girona (http://qgis.osgeo.org) and mapped for each site. Digital shape files for Unguja and Pemba administrative regions were obtained from DIVA-GIS (https://www.diva-gis.org).

## Results

### Patent schistosome infections of *Bulinus*

Over the six malacological surveys conducted on Pemba between 2016 and 2019, and the one survey conducted in Unguja, a total of 11,116 *Bulinus* spp. were collected from 65 of the 112 sites. From the subset of the snails that were identified by molecular analysis from each of these sites (n = 504), and those identified from the SCAN repository (n = 6), the majority were *B. globosus* (n = 433 snails), the remainder were *B. nasutus* (n = 77 snails). The other 10,606 *Bulinus* collected remain only morphologically identified as being within the *Bulinus africanus* species group (*Bulinus* genus), since morphological differentiation between *B. globosus* and *B. nasutus* is not possible. Of the 11,116 *Bulinus*, 0.2% (n = 20) shed *Schistosoma* spp. cercariae. These were collected from eight sites: four sites in Kinyasini (n = 16 snails), two in Kizimbani (n = 2 snails), one in Chambani (n = 1 snail) and one in Kangagani (n = 1 snail) as presented in Table 1. The infected snail from Kangagani was collected during the targeted malacological *B. nasutus* survey, in which 198 individual *Bulinus* spp. specimens were collected and at the time of collection were not shedding. Although having no observed patent schistosome

infections during the first round of shedding, a single *B. nasutus* was found shedding *Schistosoma* cercariae 21 days later. No follow up shedding was attempted on the other snails collected in Pemba as they were preserved within 48 hours of collection.

For the malacological surveys conducted at three sites in two shehias (Mtende and Kizimbani Dimbani) on Unguja, only six *B. nasutus* (Table 1) were collected in total, of which none shed *Schistosoma* cercariae within 24 hours of their collection. No further shedding attempts were made on these snails which were preserved after the first round of shedding.

### *Bulinus* spp. genetic diversity and distribution

From the subset of *Bulinus* spp. successfully sequenced, 433 out of 510 from 3 sites across Unguja and 49 sites across Pemba were *B. globosus* (S1 Table). The remaining 77 snails were identified as *B. nasutus* collected from 15 sites in Pemba and 5 in Unguja (S1 Table). *Bulinus nasutus* was molecularly identified only from freshwater bodies near the east coast of Pemba and the southern districts up to the central west areas of Unguja (Fig 2). Where >1 snail was molecularly identified per site, the snails were identified as either *B. globosus* or *B. nasutus* coming from the same site, except for from one waterbody on Pemba (Puj11, Fig 2B), where both species co-occurred in the same seasonal pond (S1 Table). Based on the subset of molecularly identified *Bulinus*, no other mixed *B. globosus* and *B. nasutus* populations were observed, however this would have to be fully confirmed with further genetic analysis of a large set of snails from each water body.

All 11 *B. globosus* and 9 out of 10 *B. nasutus* *cox*1 haplotypes were unique to either Unguja or Pemba, with one exception being *B. nasutus* haplotype 3, as shown in S2 Table, that was detected on both islands. A single clade of *B. nasutus* specimens from both Unguja and Pemba was observed, whereas the *B. globosus* isolates from each island fell into two distinct clades

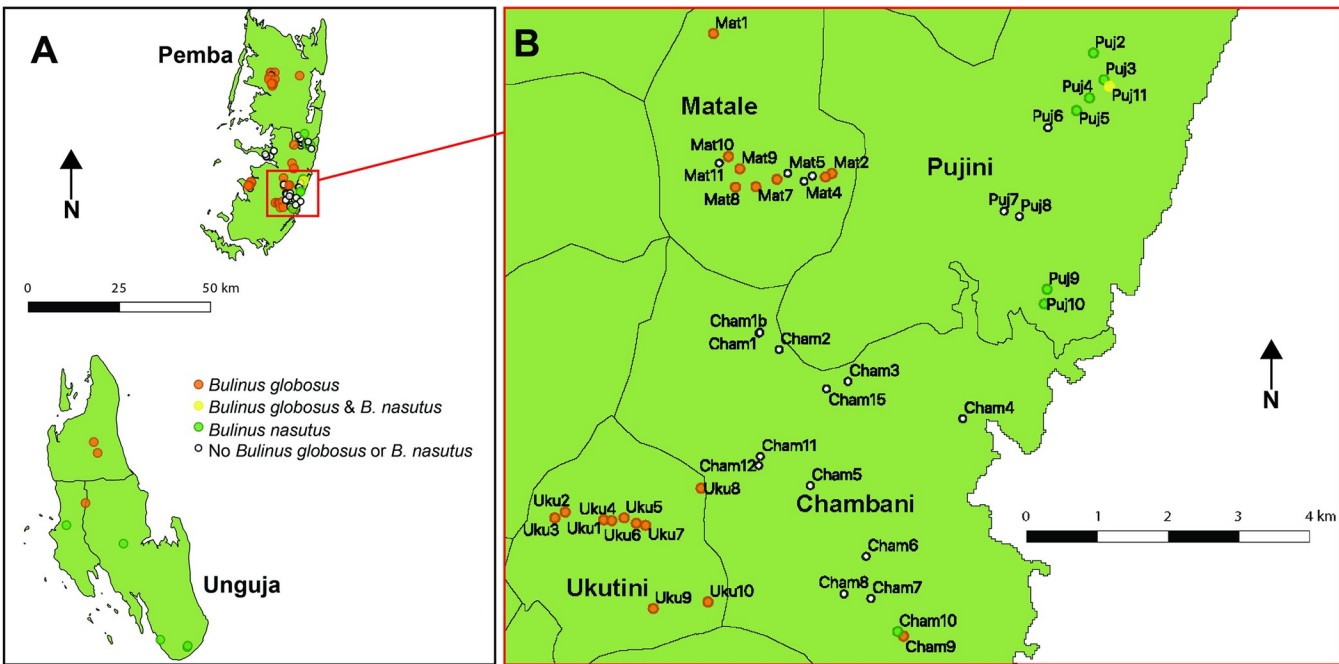

**Fig 2. A: Inferred *Bulinus globosus* and *B. nasutus* distribution on Unguja and Pemba islands (Zanzibar, United Republic of Tanzania) as identified by mitochondrial *cox*1 sequences of a subset (n = 510) of *Bulinus* spp. collected. B: Highlighted South East region of Pemba, displaying human freshwater contact sites in four shehias (Matale, Pujini, Chambani, Ukutini) and the single freshwater body cohabited by *B. globosus* and *B. nasutus* (Puj11).** Digital shape files for Unguja and Pemba administrative regions were obtained from DIVA-GIS (https://www.diva-gis.org).

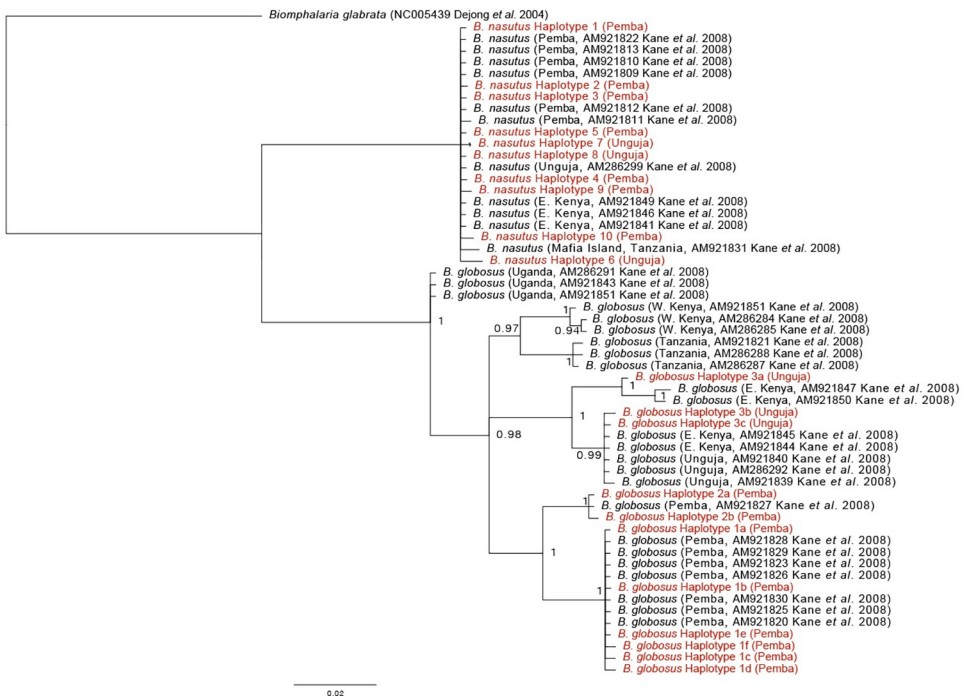

**Fig 3. Bayesian inference of the partial mitochondrial *cox*1 haplotype dataset of *Bulinus nasutus* and *B. globosus* collected from Unguja and Pemba.** Reference data from East Africa (Kane *et al.* [42]). Tree produced using Bayesian inference using MrBayes v3.2.7A [51] under the HKY+I+G model (-lnl 2020.2144, AIC 4052.4287). Branches <0.95 posterior probability collapsed. The branch length scale bar indicates the number of substitutions per site. Text in red indicates *Bulinus* haplotypes generated from the current study as listed in S2 Table.

(Fig 3). All *B. globosus* from Pemba fell into one clade containing two sister groups, with those previously identified from Pemba [42]. The three haplotypes from Unguja fell into a another clade containing two sister groups of *B. globosus* from Eastern Kenya and those previously identified from Unguja (Fig 3) [42].

## *Schistosoma* spp. cercariae identification

From the subset of cercariae identified from 89 of the infected *Bulinus* collected from Zanzibar between 2016 and 2019 (n = 20) and identified during the ZEST study (n = 69), 82 were shedding *S. haematobium* with 18 different haplotypes and seven were shedding *S. bovis* with two haplotypes (S3 and S4 Tables). Both Group 1 and 2 haplotypes of *S. haematobium* representing mainland African and Indian Ocean islands respectively were identified [27,28] (Figs 4 and S1). Including *Schistosoma* coinfections, of which there were seven determined by multiple *cox*1 haplotypes presented in S5 Table, the proportion of snails shedding *S. haematobium* Group 1 cercariae (n = 41) was similar to those shedding Group 2 cercariae (n = 45). However, the majority of Group 1 *S. haematobium* infections occurred in Unguja (n = 35), with significantly fewer (n = 6) from Pemba ($\chi^2$ = 10.0, *df* = 1, P< 0.01). In contrast, Group 2 *S. haematobium* cercariae were distributed evenly across the islands (Unguja n = 23 and Pemba n = 22). Most (n = 13 of 18) *S. haematobium cox*1 haplotypes were unique to either Pemba or Unguja, but five were present across both islands.

Seven snails as presented in S5 Table were confirmed as shedding *Schistosoma* cercariae with multiple *cox*1 haplotypes of either *S. haematobium* (n = 6) or *S. bovis* (n = 1), indicating they had been infected by multiple miracidia. Furthermore, five of the six *S. haematobium*

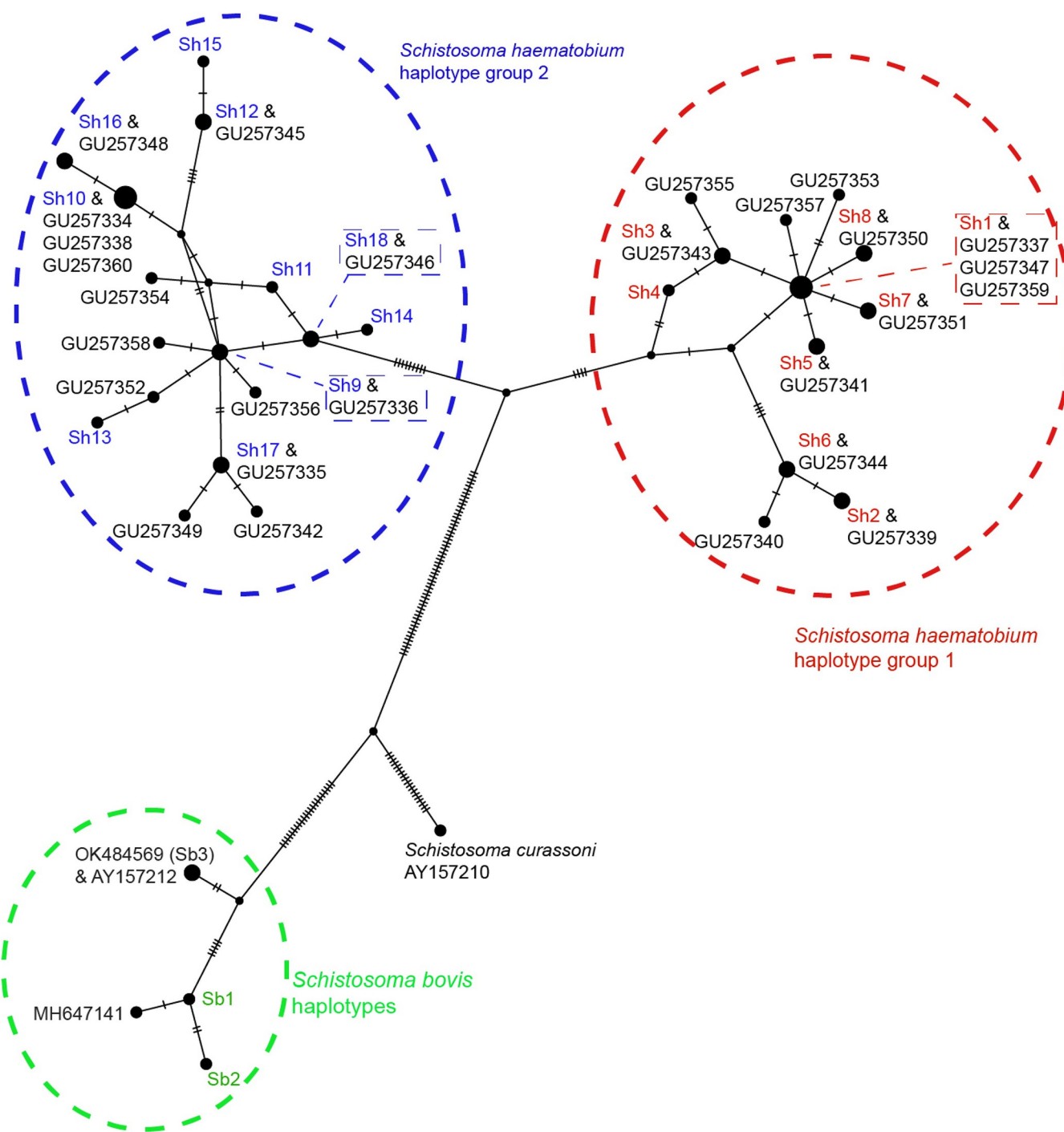

**Fig 4. TCS haplotype network of *Schistosoma* spp. partial *cox*1 DNA sequences (750 bp).** Produced using PopArt [56]. Hatches represent SNP differences from joined nodes and size of nodes is scaled to the number of identical haplotypes listed. *Schistosoma* haplotype group 1 and 2 indicates whether cercariae were identified as mainland Africa (1) or Indian Ocean Island (2) haplotypes (as described in Webster *et al*. [27]). *Schistosoma haematobium* reference haplotypes (GU257334 –GU257360) from Webster *et al*. [28]. *Schistosoma bovis* reference haplotypes (OK484569, AY157212 and MH647141) from Pennance *et al*. [54], Lockyer *et al*. [53] and Djuikwo-Teukeng *et al*. [55], respectively. *Schistosoma curassoni* reference (AY157210) from Lockyer *et al*. [53].

infected snails simultaneously shed both Group 1 and Group 2 *S. haematobium* haplotypes, whilst only one snail was identified shedding two haplotypes of Group 2.

Comparison of mitochondrial *cox*1 and nuclear ITS profiles from *S. haematobium* cercariae showed no evidence of hybridisation between *S. haematobium* and *S. bovis*. No intraspecies variation in the ITS profiles were observed, with 100% match to reference data [27].

### *Bulinus* observed shedding *Schistosoma haematobium* and *S. bovis*

Of the 20 infected snails collected in Pemba listed in S6 Table, 12 were identified as *B. globosus* infected with *S. haematobium* and seven as *B. globosus* infected with *S. bovis*. The remaining infected *Bulinus* collected during the targeted survey in Kangagani was identified as *B. nasutus* (Haplotype 3 in S2 Table), matching that previously reported on Pemba (GenBank Accession: AM921812, see [42]). The two cercariae identified from this snail were *S. haematobium* of a single *cox*1 haplotype (Sh3). This *S. haematobium* haplotype has been previously identified as 'Group 1' (GenBank Accession: GU257343, see [28]) representing those from mainland Africa and Zanzibar (Figs 4 and S1 Fig). The infected *B. globosus* from Unguja from the SCAN repository was shedding 'Group 2' cercariae as presented in S6 Table.

## Discussion

Here we update *Schistosoma* and *Bulinus* species distributions in Zanzibar as a schistosomiasis surveillance resource, while also investigating specific associations between *Bulinus* and *Schistosoma* spp. on both Pemba and Unguja islands. Of the 11,116 *Bulinus* spp. collected from Pemba and Unguja between 2016 and 2019, 0.2% were infected with *S. haematobium* group species; *B. globosus* shed *S. haematobium* (n = 12) and *S. bovis* (n = 7) and *B. nasutus* shed *S. haematobium* (n = 1). The latter host-parasite relationship is the first to confirm preliminary findings by Ame [26] whilst also refuting that *B. nasutus* is refractory to *S. haematobium* infection across Zanzibar [23]. The distribution of *B. globosus* and *B. nasutus* across Pemba and Unguja inferred from molecular identifications of a subset of those collected confirmed previous findings of separate distributions [23,29,30], with the exception that the two *Bulinus* species were present in the same waterbody at one site and in close proximity across two other neighbouring sites along the east coast of Pemba (potentially overlapping distribution) where only *B. nasutus* was known to be abundant from previous reports. Identification of a second compatible intermediate snail host for *S. haematobium* (*B. nasutus*) changes our understanding of the snail and *Schistosoma* spp. biology on Zanzibar. In addition, the confirmed presence of *B. globosus* infected with *S. bovis* almost two years following the first recording of this pathogen on the island [29] and in combination with the presence of infected cattle [54], is cause for concern since increased transmission could lead to significant animal health and economic impacts, as well as a potential risk for hybridisation with *S. haematobium* (see [59,60]).

### Distribution and diversity of *Bulinus globosus* and *B. nasutus*

Co-occurrence of *B. globosus* and *B. nasutus* was observed at just one site on Pemba (Puj11), generally supporting previous observations that species distribution in freshwater bodies is dictated by species specific ecological factors (such as water conductivity) determined by the geological zones of Zanzibar (see [61,62]). However, it is noteworthy that during the current study it was only feasible to identify a proportion (n = 510 of 11,116) of the *B. globosus* and *B. nasutus* accurately through partial *cox*1 sequencing. Therefore, it is possible that other sites containing both *B. globosus* and *B. nasutus* may be identified on the Zanzibar Archipelago in future species identification. The development of a cheap, easily interpreted, rapid diagnostic

assay to distinguish between *B. globosus* and *B. nasutus*, such as is available for the differentiation of *S. haematobium* and *S. bovis* [63], would provide a much needed solution for identifying large numbers of field collected specimens.

The 11 *B. globosus cox*1 haplotypes identified from the snails collected across Zanzibar fell into two distinct clades representing Unguja and Pemba taxa [24]. The *B. globosus* from Unguja were more closely related to those previously identified from East Kenya [42], whilst those from Pemba form an independent group distinct from the other East African isolates, suggesting independent origins. This agrees with our current understanding of Zanzibar's geological formation, whereby Pemba island separated from mainland Africa earlier, during at least the early Pliocene compared to Unguja island during the Pleistocene [61,64]. In contrast, there was no phylogenetic distinction between the *B. nasutus* from Unguja, Pemba and mainland East Africa, with samples from each region all forming a single clade of multiple haplotypes. As recorded previously, *B. globosus* is more genetically diverse than *B. nasutus* (see [65]). Identical haplotypes of *B. nasutus* were also present on Unguja and Pemba. At this stage of investigation, we postulate that since *B. nasutus* infected with *S. haematobium* observed here in Pemba matched haplotypes of *B. nasutus* in the south of Unguja (see below), it might serve as an intermediate host of urogenital schistosomiasis across the Archipelago, as is the case in Kenya [31,66] and Tanzania [32–37].

## A 'new' intermediate host of *Schistosoma haematobium* on Pemba: *Bulinus nasutus*

The finding of *B*. nasutus in one locality naturally infected with *S. haematobium* on Pemba contradicts previous results suggesting that this snail species was not involved with the transmission of urogenital schistosomiasis on Zanzibar [22–24]. Indeed, pre-patent *Schistosoma* spp. infections previously observed in *B. nasutus* from Pemba might have been *S. haematobium* (see [26]) but infections did not appear to result in cercarial production [25].

*Schistosoma haematobium* cercariae shed from the intermediate snail hosts, analysed here, fall into the two known *S. haematobium* haplotype groups (1 and 2) previously identified from miracidia collected from infected humans [27,28]. The *cox*1 haplotypes of cercariae shed from *B. nasutus* were identified as Group 1, a group predominantly associated with African mainland *S. haematobium* populations. Possibly only Group 1 *S. haematobium* is compatible with Pemba *B. nasutus*, since this same snail species acts as an intermediate host in East Africa [31–37], however more samples would be needed to test this hypothesis. *Schistosoma* strain specific interactions/compatibilities with intermediate host snails based on differential immune responses, have been well studied and established in strains of *S. mansoni* and *B. glabrata* (as summarised in [67]). However, relatively little is understood regarding co-evolution of host/parasite compatibility between *S. haematobium* group strains and *Bulinus* species, save some studies investigating geographical isolates (see [68–72]). Such strain dependency or local adaptation reflects the patchy compatibility of *Bulinus* snail hosts of *Schistosoma* generally [73], discussed in a previous study in Tanzania, where experimental infections of *B. nasutus* using a local strain of *S. haematobium* that usually infects *B. globosus* were unsuccessful [74]. As demonstrated again here, *B. globosus* remains the primary host of *S. haematobium* on Zanzibar, but endemic *B. nasutus* may also play a minor role in transmission involving specific *S. haematobium* strains, complicating future monitoring. This hypothesis also provides some explanation to how 'intermittent' and/or 'unstable' transmission was historically maintained in areas of Zanzibar (such as south Unguja) where only *B. nasutus* is present currently [7,9,10,75,76].

### Study limitations and future work

Several limitations are apparent in the current study. The time and costs required to generate *cox*1 sequence data limited the number of snail intermediate hosts that could be identified by this means. Development and testing of a rapid diagnostic assay, or alternative, to provide rapid high throughput identification of snails would greatly support future studies. Additionally, few *Bulinus* specimens were available for analysis from Unguja, so further malacological surveys with molecular sub-sampling here would be beneficial to confirm snail species distributions across the island. Also, since the majority of infected *Bulinus* spp. collected during the ZEST studies were not accessioned with their associated *Schistosoma* spp., complete inferences on snail-*Schistosoma* relationships were not possible. Finally, although cercariae identification was used here to identify *Schistosoma* species, it would have been of interest to identify a greater number of cercariae per snail infection, as this may have significantly increased the number of coinfections observed from these *Bulinus* spp. and allow for the potentially immune modulated interactions of Group 1 and Group 2 *S. haematobium* coinfections, observed in five snails here, to be explored further. In a future study, it would also be of interest to use PCR based methods to identify pre-patent *S. haematobium* and *S. bovis* infections in snails to further assess the total number of snails that have been exposed to schistosomes but are not currently contributing to transmission [77].

### Implications for future monitoring of schistosomiasis on Zanzibar

The findings discussed here provide implications for future control and elimination efforts of urogenital schistosomiasis on Zanzibar [18,19]. First, it is suggested here that the Ministry of Health, Social Welfare, Elderly, Gender and Children Zanzibar should conduct periodic surveys in areas associated with *B. nasutus* distribution on Unguja and Pemba. These surveys should include both malacological collections and combined human urine collections with questionnaires (including questions on freshwater usage locally and elsewhere in Zanzibar), followed by targeted treatment of infected individuals and focal snail control, to reduce any ongoing transmission. Second, a monitoring system to check the identity of schistosome cercariae shed from *Bulinus* spp. snails from Zanzibar to differentiate bovine and human schistosomiasis would enable accurate mapping of both schistosome species, appropriate targeting of urogenital schistosomiasis control interventions, and also monitor any potential hybridization events between *S. haematobium* and *S. bovis* that may be occurring in cattle or humans [60].

Nearly a century has passed since the first reports of widespread human urogenital schistosomiasis on Zanzibar, during which time there have been great achievements towards urogenital schistosomiasis elimination. As the battle to eliminate schistosomiasis from the Zanzibar Archipelago continues, our findings emphasize the need to carefully plan future surveillance strategies of transmission on the islands, taking into consideration the presence of bovine *Schistosoma* species and the capacity for an expanded intermediate host, and therefore geographical, range of *S. haematobium*.

### Supporting information

**S1 Table. Associated *Bulinus* spp. specimen data.** Detailed information on *Bulinus* spp. specimens used in current study, including species identification determined through molecular analysis (*cox*1), collection site name including latitude and longitude, schistosome species patency and *cox*1 haplotype.
(XLSX)

**S2 Table. *Bulinus* spp. *cox*1 haplotypes observed from Pemba and Unguja, and associated *Schistosoma* spp. infecting each snail haplotype.** [a] *Schistosoma* haplotype group indicates whether cercariae were identified as mainland Africa (1) or Indian Ocean Island (2) haplotypes (as described in Webster *et al.* [27]). [b] It was not possible to associate this snail haplotype with its *S. haematobium* cercariae *cox*1 haplotype(s) as it was preserved with two other *S. haematobium* infected *B. globosus*.
(XLSX)

**S3 Table. Associated *Schistosoma* spp. specimen data.** Detailed information on *Schistosoma* spp. specimens used in current study, including species identification determined through molecular analysis (*cox*1 and ITS1-5.8S-ITS2), collection site name including latitude and longitude and *Schistosoma cox*1 haplotype.
(XLSX)

**S4 Table. *Schistosoma cox*1 haplotypes identified from cercariae from Unguja and Pemba.** [a] *Schistosoma* haplotype group indicates whether cercariae were identified as mainland Africa (1) or Indian Ocean Island (2) *cox*1 haplotypes (as described in Webster *et al.* [27]). [b] Island; U = Unguja, P = Pemba.
(XLSX)

**S5 Table. Trematode coinfections of *Bulinus* spp. from Unguja and Pemba.** [a] *Schistosoma* haplotype group indicates whether cercariae were identified as mainland Africa (1) or Indian Ocean Island (2) *cox*1 haplotypes (as described in Webster *et al.* [27]). [b] Coinfection indicates the *Schistosoma cox*1 haplotype group (as described in [27]) infection profile of each *Bulinus* spp. Sh1 = *Schistosoma haematobium* mainland African haplotype group 1, Sh2 = *S. haematobium* Indian Ocean haplotype group 2, Sb1 = *S. bovis* haplotype 1, Sb2 = *S. bovis* haplotype 2.
(XLSX)

**S6 Table. *Bulinus* spp. infected with *Schistosoma* spp. identified from Unguja and Pemba.** [a] *Schistosoma* haplotype group indicates whether cercariae were identified as mainland Africa (1) or Indian Ocean Island (2) haplotypes (as described in Webster *et al.* [27]).
(XLSX)

**S1 Fig. Bayesian inference of the partial mitochondrial *cox*1 haplotype dataset of *Schistosoma haematobium* and *S. bovis* collected from Unguja and Pemba.** Phylogenetic tree constructed using Bayesian inference in MrBayes v3.2.7a [51] under the HKY + I model (-lnL = 1809.8890, AIC 3629.7781, ASDOSF < 0.01 at 1,791,000 generations). Branches with <0.95 posterior probability are collapsed. The branch length scale bar indicates the number of substitutions per site. Text in red indicates *Schistosoma* haplotypes generated in the current study.
(TIF)

## Acknowledgments

We are grateful for the local residents of Unguja and Pemba, for their help in locating freshwater bodies for malacological surveys and providing other insightful comments that has shaped this work. We thank Prof. Russell Stothard (Liverpool School for Tropical Medicine) for providing snail specimens and adding to the Schistosomiasis Collection at the Natural History Museum used in this study, and for his useful discussions on the subject of schistosomiasis transmission in East Africa.

## Author Contributions

**Conceptualization:** Tom Pennance, Joanne Cable, Bonnie L. Webster.

**Data curation:** Tom Pennance, John Archer, Muriel Rabone, Stefanie Knopp, Bonnie L. Webster.

**Formal analysis:** Tom Pennance, John Archer, Stefanie Knopp, Bonnie L. Webster.

**Funding acquisition:** Tom Pennance, Shaali Makame Ame, Fatma Kabole, Said Mohammed Ali, Stefanie Knopp, David Rollinson, Joanne Cable, Bonnie L. Webster.

**Investigation:** Tom Pennance, Shaali Makame Ame, Amour Khamis Amour, Khamis Rashid Suleiman, Mtumweni Ali Muhsin, John Archer, Stefanie Knopp, Joanne Cable, Bonnie L. Webster.

**Methodology:** Tom Pennance, Shaali Makame Ame, Stefanie Knopp, Bonnie L. Webster.

**Project administration:** Tom Pennance, Shaali Makame Ame, Amour Khamis Amour, Khamis Rashid Suleiman, Mtumweni Ali Muhsin, Fatma Kabole, Said Mohammed Ali, Stefanie Knopp, Bonnie L. Webster.

**Resources:** Tom Pennance, Shaali Makame Ame.

**Supervision:** Shaali Makame Ame, Fatma Kabole, Said Mohammed Ali, Fiona Allan, Aidan Emery, Stefanie Knopp, David Rollinson, Joanne Cable, Bonnie L. Webster.

**Validation:** Tom Pennance.

**Visualization:** Tom Pennance, Bonnie L. Webster.

**Writing – original draft:** Tom Pennance.

**Writing – review & editing:** Tom Pennance, Shaali Makame Ame, John Archer, Fiona Allan, Aidan Emery, Muriel Rabone, Stefanie Knopp, David Rollinson, Joanne Cable, Bonnie L. Webster.

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
