## [Decision Letter · Decision Letter 0]

6 Mar 2022

Dear Dr. Pennance,

Thank you very much for submitting your manuscript "Transmission and diversity of Schistosoma haematobium and S. bovis and their freshwater intermediate snail hosts Bulinus globosus and B. nasutus in Zanzibar" for consideration at PLOS Neglected Tropical Diseases. As with all papers reviewed by the journal, your manuscript was reviewed by members of the editorial board and by several independent reviewers. In light of the reviews (below this email), we would like to invite the resubmission of a significantly-revised version that takes into account the reviewers' comments. 

All three reviewers found the submitted paper useful and addressing an important topic. Nevertheless, they have raised some significant points, particularly with regard to identification of the snails and presentation of some of the data, which should be the focus for preparation of any resubmission.

We cannot make any decision about publication until we have seen the revised manuscript and your response to the reviewers' comments. Your revised manuscript is also likely to be sent to reviewers for further evaluation.

Sincerely,

Stephen W. Attwood, BSc,MSc,PhD

Associate Editor

Simone Haeberlein, PhD

Deputy Editor

Reviewer's Responses to Questions

**Key Review Criteria Required for Acceptance?**

**Methods**

-Are the objectives of the study clearly articulated with a clear testable hypothesis stated?

-Is the study design appropriate to address the stated objectives?

-Is the population clearly described and appropriate for the hypothesis being tested?

-Is the sample size sufficient to ensure adequate power to address the hypothesis being tested?

-Were correct statistical analysis used to support conclusions?

-Are there concerns about ethical or regulatory requirements being met?

Reviewer #1: See below

Reviewer #2: Overall, the methods employed seem reasonably sound. However I have some serious concerns regarding the number of snails and cercariae examined in the study.

My principal concern relates to the identification of the snails. Table 1 shows that a very large number of snails have been collected in the study but that only a very small number of these snails have been identified using molecular methods. As I initially read the methods section I assumed (as it turns out incorrectly) that all snails collected in the study had been identified (presumably using morphological methods) with molecular methods used to confirm identifications for just a small number of individuals. Such confirmation of identifications would have been perfectly acceptable but it turns out that the authors never actually bothered to identify the vast majority of snails they collected. Instead, we are told at the start of the results that the snails in a population were 'retrospectively' assigned as B. globosus or B nasutus based on the very limited number of samples for which molecular identifications were undertaken. So for the first site shown in Table 1 (Ukutine 21/2/2018) it is implied that 750 B. globosus snails were collected at that site, yet only 29 snails were actually identified as B. globosus. We see the same issue at all other sites. There would seem to be little point collecting such large numbers of snails from a site if you don't bother to look at them (to identify them) and it is most definitely not appropriate to 'retrospectively' identify snails in this way and in effect mislead the reader by implying that considerably more work has been undertaken than has actually been done. 

The authors should therefore only present the data that they have. So for the site Ukutine in 21/2/2018, they collected 29 B. globosus specimens and B. globosus made up 100% of the snails identified at that site. Indeed this is exactly how most field studies are undertaken. We sample 20-30 individuals from a site which we identify and based on the proportions in our sample we determine the percentage of each species present. We don't imply that we have data for 750 snails when we do not.

Unfortunately for the authors, while sites like Ukutine with 29 identified specimens have been sampled reasonably well, other sites have not. Kizimbani 2016 has just 6 samples, Wawi 2016 has just 3 samples. The fact that you collected 199 snails from Kizimbani in 2016 and 149 snails from Wawi in 2016 is completely irrelevant as you didn't bother to look at them. Unfortunately there are many other sites with such pitiful numbers of identified snails.

The next question is does this matter i.e. is this likely to impact the results. The short answer is yes it does matter. Unfortunately, it is not the case that you only ever find monomorphic populations (with just a single species at a site) so sampling only a handful of individuals and inferring that this applies to the site as a whole is not enough. The authors own data in Table 1 shows that two sites have mixed populations of B. globosus and B. nasutus (Pujini 22/7/2018 and Chambani 24/10/2016). So for a site like Wawi with just 3 individuals identified and a site like Kizimbai with 6 individuals I have no confidence whatsoever that these populations are purely B. globosus and I wonder how many additional mixed populations might have been found if proper sample identification had been undertaken. I would suggest that a minimum of 20 individuals per site per time point would need to be identified for confidence.

As well as my concerns re snail identifications, I also have some concerns over the number of cercariae samples. Rather than tell us in the methods that a subset were sampled please be clear and tell us exactly how many cercariae were sampled. I do also wonder why the authors have used cercarial shedding to determine infection rate rather than use PCR based methods of infection detection. Such PCR based methods are much simpler and avoid inaccuracy due to missed pre-patent individuals (though having said this the authors do make an effort to screen for cercaria on more than one occasion; at least for the majority of populations).

Reviewer #3: (See below)

**Results**

-Does the analysis presented match the analysis plan?

-Are the results clearly and completely presented?

-Are the figures (Tables, Images) of sufficient quality for clarity?

Reviewer #1: See below

Reviewer #2: My principal criticism of the results relates to the 'retrospective identification' of snail samples (or rather guessing the identification of the population at large based on a very small inadequate number of samples identified). Please see my comments above in the methods section of this review. I will not repeat these comments here but in essence the poor sampling undertaken (very few snails identified for some sites) renders all conclusions re presence or absence of B. globosus/ B. nasutus unsound.

Paraphyletic would seem inappropriate when referring to the clades in the tree. Remove and just refer to them as clades.

Please also note that the results text does not seem to match the data in Table 1. In the text we are told that just 1 site has a mixed population of B. globosus and B. nasutus. In Table 1, two sites are shown to have a mixed population of B. globosus and B. nasutus. I would also like to see some number in Table 1 for these 2 sites. How many of the individuals identified were B. globosus and how many B. nasutus.

Reviewer #3: (See below)

**Conclusions**

-Are the conclusions supported by the data presented?

-Are the limitations of analysis clearly described?

-Do the authors discuss how these data can be helpful to advance our understanding of the topic under study?

-Is public health relevance addressed?

Reviewer #1: See below

Reviewer #2: Certainly some of the conclusions are not supported by the data presented. I have no confidence in conclusions relating to the presence/absence of B.globosus/B. nasutus at collection sites due to the inadequate number of snails identified at many of these sites.

However, I have confidence in the conclusion that B. nasutus acts as an intermediate host for S.haemotobium and is thus an important consideration in schistosomiasis control in humans.

Reviewer #3: (See below)

**Editorial and Data Presentation Modifications?**

Reviewer #1: See below

Reviewer #2: It is stated in the manuscript that Schistosomiasis transmission on Zanzibar was believed to involve a single schistosome species (Schistosoma haematobium) transmitted via a single intermediate host species (Bulinus globosus). Statements to ths effect are not accurate and should be removed from the manuscript. I do appreciate that the authors are keen to enhance the impact of their work in this paper but S.bovis has previously been reported on Zanzibar by the authors of this paper in a previous paper from 2018. The discovery of S. bovis on Zanzibar is thus not a discovery new to this paper and should not be presented as such.

Reviewer #3: (See below)

**Summary and General Comments**

Reviewer #1: The paper is interesting and useful. There are some details that need to be attended to before publication. 

My main point of confusion concerns the number of infected snails. In the abstract and in line 364, 89 snails were said to be infected. This probably matches the number in Table S2, but I haven’t checked in detail. In other places in the paper, it is stated that 20 snails were infected (line 307). In Table 1 and Table S1, I can identify 22 infected snails. In Table 5, there seem to be 21 snails. I assume that the 69 unidentified snails are those referred to in line 251. These numbers need to be clarified.

There are some points of English expression and other details that also caught my attention.

Line 33. “… allopatric for S. haematobium…” What does that mean?

Line 48 and elsewhere. Which islands are regarded as being members of the Indian Ocean Islands?

Line 52. “…evidence that intraspecific variability may increase the transmission potential…”. Th evidence is not very convincing to me.

Lines 81-83. Low-level transmission maintained through exposure to cercariae released from infected snails. This seems to be stating the obvious. What point is being made?

Line 171 and many other places. I would not personally use “represents”. Maybe “indicates”?

Lines 181-182. “…searching as much of the perimeter of the water body as possible…”?

Lines 207-209. This sentence should be rephrased or split.

Lines 238-241. As far as I can tell, you sequenced DNA from 510 snails, not the 1009 snails as implied here. Maybe clarify that here

Line 248. The cercariae were apparently processed individually, not pooled. Perhaps make that clear.

Lines 324, 418 and maybe elsewhere. Don’t start a sentence with an abbreviated genus name.

Line 336. Paraphyletic with respect to what? In each case, they look like sister clades to me. Please check that the rest of this description of the tree in Fig. 3 is accurate.

Line 368. “co-infections” here, with a hyphen. Without a hyphen elsewhere.

Fig. 2. At least in the review pdf, the symbols on figures are rather small and hard to read. The coloured circles in Fig. 2 could be made larger for a start. In Fig. 4, the lettering is too small.

Is Fig 5 necessary, given Fig. 4? Also, inclusion of both species separated by one very long branch has the effect of rendering the branches within S. haematobium (in particular) too short to be visualised properly. Indeed, Group 1 and Group 2 haplotypes seem to be thoroughly intermingled in this tree.

Line 459. Two distinct clades – I agree. But each of these seems to consist of two distinct subclades. Might that have some significance?

Line 494. Local strain of…

Reviewer #2: There is some good data here and the finding that B.nasutus acts as an intermediate host of S. haematobium is of particular interest. Unfortunately, this very interesting finding is overshadowed by the inadequate number of snails identified at many of the collection sites. It is simply not safe to conclude that just a single species of snail is present at a site when only a handful of snails have been identified. The findings relating to the presence/absence of B.globosus/B.nasutus at sites on Zanzibar therefore cannot be trusted. I therefore do not consider that the paper can be published in its present form. 

However, I do note that the authors have undertaken excellent sample collections from Zanzibar with large numbers of snails collected from many sites. I would therefore encourage the authors to revisit their data and to undertake identifications of an adequate number of snails (minimum 20 per site per time point) and resubmit.

Reviewer #3: (See below)

PLOS authors have the option to publish the peer review history of their article (what does this mean?). If published, this will include your full peer review and any attached files.

Reviewer #1: No

Reviewer #2: No

Reviewer #3: Yes

Reviewer #3: Comments

This manuscript is a sequel to previous articles and a project named Zanzibar

Elimination of Schistosomiasis Transmission (ZEST) project (2012-2017) that aimed at

eliminating schistosomiasis from the Zanzibar Archipelago. The authors collected three

species of Bulinus snails from the Zanzibar Archipelago in the United Republic of

Tanzania between 2016 and 2019. Snail were exposed to cercarial emergence to identify

Schistosoma in these islands. Snails were infected with two Schistosoma species: S.

haematobium (n=82) and S. bovis (n=7). Phylogenetic analysis of the mitochondrial

COI on 433 individuals of Bulinus globosus and 77 individuals of Bulinus nasutus

revealed three distinct haplotypes of B. globosus and one haplotype of B. nasutus. The

authors also analyzed a partial mitochondrial COI region and the nuclear ITS rDNA

region of 179 Schistosoma cercariae, identifying 18 haplotypes of S. haematobium and

three haplotypes of S. bovis. I found this manuscript original and relevant to the

parasitological and malacological field since it provides more precise information aimed

at a better understanding of the risk of schistosomiasis in Tanzania.

I found the manuscript very interesting for those with closely related research interests:

parasitologists studying zoonotic pathogens in Tanzania and other Africa countries, for

instance. The methodology is appropriate. I am not a specialist in schistosomiasis nor in

Bulinus snails but I think that references are fine. I found one major issue, however, that

should be ameliorated to make the manuscript more attractive and easier to follow by

the reader. Results should be shorten and presented more straightforward to make the

manuscript clearer. This issue can be easily achieved with some rewriting. The way the

authors present the results (text, figures and tables) is confusing or too complicated (see

below). I would reduce the number of figures and tables. I would delete some tables and

figures and modify others (see below). I am not sure if the journal allow supplementary

material. If it does, I would move some figures to supplementary material (see comment

below). There are also some confusing sentences or incorrect assumptions. For

example, I think that these facts should not be related (see below): “Although S.

haematobium on Zanzibar is genetically more diverse in comparison to mainland

Africa, with the presence of both Group 1 (mainland Africa) and the more diverse

Group 2 (Indian Ocean Islands) strains, it remained the only Schistosoma species

identified on the islands”.

There are two other major issues that should be emended. First, the authors say the

manuscript demonstrates that the snail species B. nasutus could play a minor role in

transmitting Schistosoma in Zanzibar because they found one individual infected with

the parasite. However, I think that the fact that suggest that this snail species could be

playing a role in transmitting the parasite is not having found one infected snail but,

instead, the fact that this snail species transmits the disease in Kenya and Tanzania.

There is no reason to believe, I think, that if the snail species is present in the

archipelago, it would not transmit the disease. This fact should clarify because it is very

confusing as it is.

Second, the authors infer that all the snail individuals found in a given site belong to the

species identified by sequencing 1-29 individuals. In my view, this is a very risky (and

wrong) assumption and should be avoided. It is very likely that some of the sampled

sites have more than one Bulinus species. In fact, the authors found two species in one

of the sites. The authors should clarify the number of snail individuals that have been

identified at the species level. They could say that XX individuals have been identify at

the species level while XXX individuals remain only identified at the genus level. Theycould mention in the Discussion section that communities with more than one Bulinus species are common.

Here I made some other comments:

Title:

o It should better specify the geographic area studied. Include “Zanzibar

Archipelago, Tanzania” in the title.

Abstracts

o Delete the following phrase “The involvement of B. nasutus (a snail

species that occupies geographically distinct regions on the Archipelago)

in S. haematobium transmission has previously been debated.”. It is the

second phrase in the abstract and is very confusing. It does not saying

anything as it is.

o Line 52:host or parasite intraspecific variability?

o Line 53: “intraspecific variability may increase the transmission potential

of S. haeamtobium due to the expanded intermediate host range”. I think

that both facts are not necessarily linked.

o The total number of collected snails and the number of sampled sites

should be present in the abstracts since it is one of the highlights of the

manuscript. There is a great sample effort and that should be

emphasized.

Introduction:

o Line 105: Snail habitat preferences?

o Line 106-109: “Although S. haematobium on Zanzibar is genetically

more diverse in comparison to mainland Africa, with the presence of

both Group 1 (mainland Africa) and the more diverse Group 2 (Indian

Ocean Islands) strains, it remained the only Schistosoma species

identified on the islands”. I think that both facts are independent. In fact,

the authors say latter that this is not true: S. haematobium is genetically

diverse in the islands and it is not the only Schistosoma species present

in this area.

o Figure 1: I think this figure is relevant but I would midify it and include

it in the Result section. A figure that compares previous findings by

Stothard et al. [23], Pennance et al. [28,30] and Knopp et al. [18] with

the results from this manuscript would be more useful. The reader would

better visualize which are the inputs of this manuscript. The authors

could highlight the regions where the snail and parasite species have

been found and put dots in the localities from this study where the

authors have found snail and parasite. The manuscript would be become

clearer and the figure would be more helpful to the reader.

Material and methods

o Table 1: this table is very extensive for the main text. I would include it

in Supp. Mat. In the main text, the authors should include, however, an

abstract of this table mention the total number of sampled snails, the total

number analyzed from the museum collection, the total number of

molecularly identified snails, etc. It is very difficult to find in the

manuscript this information? How many snails have been collected?

How many snails have molecularly analyzed?•

•

o Line 178: “snails were identified using shell morphology to their

genera”. The authors should clarify that species cannot be

morphologically identified at species level. And that thus why molecular

analysis are needed.

o Line 205: “after being stored for at least 48 hours in transparent 15L

water containers (allowing for any sediment to settle and eliminate the

risk of introducing live schistosome eggs, miracidia or cercariae into the

aquaria)”. I am not an expert in Schistosoma. Cannot eggs live more than

48H?

o Line 242: “before being collapsed into cox1 haplotype groups.” Which

software was used for identifying haplotypes?

o Line 247: Again, the total number of individuals analyzed is not clear. It

is very difficult to infer how many cercariae per snail have been

analyzed? DNA was extracted individually? Or DNA from cercariae

belonging to one infected snail were collectively extracted?

Results

o Line 302: I think that this assumption is incorrect (see comment above):

“from that site were restrospectively assigned as B. globosus or B.

nasutus based on these identifications.”.

o Line 301: “a total of 11,110 B. globosus and B. nasutus were collected

from ...” What about the third undescribed species?

o Line 324. Replace “B. nasutus” by “Bulinus nasutus”.

o Line 326: Again I think that this assumption in not correct: “Where >1

snail was identified per site, each site was either inhabited solely by B.

globosus or B. nasutus”.

o Figure 2: I would combine Figures 1 and 2. Se comment above.

o I would move Tables 2, 3, 4 and 5 to Supp. Mat. I would present instead

a single table that resume the results of the manuscript. These tables are

very descriptive as they are but I think the reader would need some table

that resume and illustrate the results of the manuscript.

o I would move Figures 3 and 5 to Supp. Mat.

o Line 366: Replace “S2 Table” by “Table S2”.

o Figure 4 is very informative. I really like it.

Discussion

o Replace 0.18% by 0.2%. Decimals in prevalence should be only used

when prevalence is lower than 1% and higher than 99%. See

http://adc.bmj.com/content/100/7/608

o Please, explain the following idea mentioned in the Abstract:

“preliminary evidence that intraspecific variability may increase the

transmission potential of S. haematobium”. I found it relevant and it

should be better explained in the Discussion section. Why the authors

believe that intraspecific variability may increase the transmission of a

disease?

Figures and tables:

This version of the manuscript has 5 figures and 5 tables. I think it too much. I

think that two figures (one showing a map with previous and current results and

other showing the parasite haplotypes) and a table resuming the manuscript

results are more than enough for such a manuscript.•

Avoid using symbols (*, a , b ...) in the caption section. It is confusing and it

takes time to understand what the authors meant. Try to include these remarks in

the caption. For instance, in Figure 1, the sentences “* Bulinus spp. distribution

inferred from Stothard et al. [23] and Pennance et al. [28,30].” and “φ

Schistosoma haematobium infection distribution interpreted from Knopp et al.

[18].” could directly appear in the caption without including the symbols.

###### End of Reviewer 3 comments.
---

## [Decision Letter · Decision Letter 1]

3 Jun 2022

Dear Dr. Pennance,

Thank you very much for submitting your manuscript "Transmission and diversity of Schistosoma haematobium and S. bovis and their freshwater intermediate snail hosts Bulinus globosus and B. nasutus in the Zanzibar Archipelago, United Republic of Tanzania" for consideration at PLOS Neglected Tropical Diseases. 

We thank you for revising the manuscript. There were some more points raised by one reviewer that will further enhance the impact and readability of your work. We are likely to accept this manuscript for publication, providing that you modify the manuscript according to the review recommendations. 

Sincerely,

Stephen W. Attwood, BSc,MSc,PhD

Associate Editor

Simone Haeberlein, PhD

Deputy Editor

Reviewer's Responses to Questions

Summary and General Comments

Reviewer #1: The paper is getting closer to being acceptable and will be a useful addition to the literature. I have few quibbles with the data but would like to appeal for greater clarity in the writing. 

Line 33. My original question about “allopatric” still stands. In my opinion, the term is simply mis-used by the authors. Surely they mean that until recently S. haematobium was thought to be the only Schistosoma species on the archipelago, but that transmission of S. bovis has now been demonstrated there.

Line 52. Again, Indian-Ocean Islands. I agree that there are many of these, but presumably the authors know which ones they mean. Are these the Zanzibar Archipelago and Mafia? Or does that number include Mauritius and Madagascar? Anywhere else? Clarity please.

Lines 88-89. Just say that low-level transmission is continuing. Many more words than required are used here.

Line 116. Maybe an apostrophe after “species”? 

Line 117. Clumsy phrasing “freshwater waterbodies”. Maybe “freshwater habitats” or something else.

Line 119. “overlapping with B. globosus”.

Line 149. “…between 2011 and 2019”. But 2016 is the year given elsewhere.

Sentence starting on line 320. Rephrase to remove the repetition of “remain(ing)”.

Line 343. Are you implying that, where >1 snail was identified at a site, all other snails from that site were regarded as belonging to that species? The phrasing is poor.

Line 347. This seems to be a place where “cohabiting” could be replaced with “sympatric”.

Lines 379-382. This sentence should be split and rephrased for clarity. As it stands, the subject of the sentence seems to be cercariae, but the authors are then clearly referring to snails.

Lines 398-399. Maybe “…variants other than those previously sequenced...“

Line 428. Where are the risk maps?

Line 434. “negating” is the wrong word. Maybe “refuting”.

Line 452. Spelling of “conductivity”.

Line 456. Be consistent in use (or not) of am initial capital letter in “archipelago”.

Lines 506-509. Use fewer words. “ the cost and time required to generate cox1 sequence data limited the number of the snails that could be identified by this means”. Or something similar.

Reviewer #3: I would like to thanks the authors for considering my comments and those of my colleagues. I found that the manuscript is much clearer and easier to follow as it is now presented.

Figure Files:

Data Requirements:

Reproducibility:

References

---

## [Editor Report · Decision Letter 2]

14 Jun 2022

Dear Dr. Pennance,

We are pleased to inform you that your manuscript 'Transmission and diversity of Schistosoma haematobium and S. bovis and their freshwater intermediate snail hosts Bulinus globosus and B. nasutus in the Zanzibar Archipelago, United Republic of Tanzania' has been provisionally accepted for publication in PLOS Neglected Tropical Diseases.

Best regards,

Stephen W. Attwood, BSc,MSc,PhD

Associate Editor

Simone Haeberlein, PhD

Deputy Editor

---

## [Editor Report · Acceptance letter]

30 Jun 2022

Dear Dr. Pennance,

We are delighted to inform you that your manuscript, "Transmission and diversity of Schistosoma haematobium and S. bovis and their freshwater intermediate snail hosts Bulinus globosus and B. nasutus in the Zanzibar Archipelago, United Republic of Tanzania," has been formally accepted for publication in PLOS Neglected Tropical Diseases.

Best regards,

Shaden Kamhawi

co-Editor-in-Chief

Paul Brindley

co-Editor-in-Chief
